# Bayesian Nonparametric Spectral Estimation

**Felipe Tobar**
Universidad de Chile
`ftobar@dim.uchile.cl`

## Abstract

Spectral estimation (SE) aims to identify how the energy of a signal (e.g., a time series) is distributed across different frequencies. This can become particularly challenging when only partial and noisy observations of the signal are available, where current methods fail to handle uncertainty appropriately. In this context, we propose a joint probabilistic model for signals, observations and spectra, where SE is addressed as an exact inference problem. Assuming a Gaussian process prior over the signal, we apply Bayes' rule to find the analytic posterior distribution of the spectrum given a set of observations. Besides its expressiveness and natural account of spectral uncertainty, the proposed model also provides a functional-form representation of the power spectral density, which can be optimised efficiently. Comparison with previous approaches, in particular against Lomb-Scargle, is addressed theoretically and also experimentally in three different scenarios. Code and demo available at `github.com/GAMES-UChile`.

## 1 Introduction

The need for frequency representation arises naturally in a number of disciplines such as natural sound processing [1, 2], astrophysics [3], biomedical engineering [4] and Doppler-radar data analysis [5]. When the signal of interest is known without uncertainty, the frequency representation can be obtained by means of the Fourier transform [6]. However, real-world applications usually only provide us with a limited number of observations corrupted by noise. In this sense, the main challenge in Spectral Estimation (SE) comes from the fact that, due to the convolutional structure of the Fourier transform, the uncertainty related to missing, noisy and unevenly-sampled data propagates across the entire frequency domain. In this article, we take a probabilistic perspective to SE, thus aiming to quantify uncertainty in a principled manner.

Classical—yet still widely used—methods for spectral estimation can be divided in two categories. First, parametric models that impose a deterministic structure on the latent signal, which result in a parametric form for the spectrum [7–9]. Second, nonparametric models that do not assume structure in the data, such as the periodogram [10] computed through the Fast Fourier Transform (FFT) [11]. Uncertainty is not inherently accounted for in either of these approaches, although one can equip parameter estimates with error bars in the first case, or consider subsets of training data to then average over the estimated spectra.

Despite the key role of the frequency representation in various applications as well as recent advances in probabilistic modelling, the Bayesian machinery has not been fully exploited for the construction of rigorous and meaningful SE methods. In particular, our hypothesis is that Bayesian nonparametric models can greatly advance SE theory and practice by incorporating temporal-structure parameter-free generative models, inherent uncertainty representation, and a natural treatment of missing and noisy observations. Our main contribution is then to propose a nonparametric joint generative model for a signal and its spectrum, where SE is addressed by solving an exact inference problem.

## 2 Background

### 2.1 Prior art, current pitfalls and desiderata

The beginning of a principled probabilistic treatment of the spectral estimation problem can be attributed to E.T. Jaynes, who derived the discrete Fourier transform using Bayesian inference [12]. Then, G.L. Bretthorst proposed to place a prior distribution over spectra and update it in the light of observed temporal data, for different time series models [13]. This novel approach, in the words of P.C. Gregory, meant a *Bayesian revolution in spectral analysis* [14]. The so developed conceptual framework paved the way for a plethora of methods addressing spectral estimation as (parametric) Bayesian inference. In this context, by choosing a parametric model for time series with closed-form Fourier transform, a Bayesian treatment provides error bars on the parameters of such a model and, consequently, error bars on the parametric spectral representation, e.g., [15–17].

Within Bayesian nonparametrics, the increasing popularity and ease of use of Gaussian processes (GP, [18]), enabled [19, 20] to detect periodicities in time series by (i) fitting a GP to the observed data, and then (ii) analysing the so learnt covariance kernel, or equivalently, its power spectral density (PSD). Although meaningful and novel, this GP-based method has a conceptual limitation when it comes to nonparametric modelling: though a nonparametric model is chosen for the time series, the model for the PSD (or kernel) is still only parametric. Bayesian nonparametric models for PSDs can be traced back to [21], which constructed a prior directly on PSDs using Bernstein polynomials and a Dirichlet process, and more recently to [22, 23], which placed a prior on covariance kernels by convolving a GP with itself. Yet novel, both these methodologies produced intractable posteriors for the PSDs, where the former relied on Monte Carlo methods and the latter on variational approximations.

The open literature is lacking a framework for spectral estimation that is:

- Nonparametric, thus its complexity grows with the amount of data.
- Bayesian, meaning that it accounts for its own uncertainty.
- Tractable, providing exact solutions at low computational complexity.

We aim to fulfil these desiderata by modelling time series and their spectra, i.e., Fourier transform, using Gaussian processes. A key consequence of using GPs is that missing/unevenly-sampled observations are naturally handled.

### 2.2 The Fourier transform

Let us consider a signal, e.g., a time series or an image, defined by the function $f : \mathcal{X} \mapsto \mathbb{R}$, where for simplicity we will assume $\mathcal{X} = \mathbb{R}$. The spectrum of $f(t)$ is given by its Fourier transform [6]

$$F(\xi) = \mathcal{F}\{f\}(\xi) \triangleq \int_{\mathcal{X}} f(t) e^{-j2\pi\xi t} \mathrm{d}t \tag{1}$$

where $j$ is the imaginary unit and the frequency $\xi$ is the argument of the function $F(\cdot)$. Notice that for $F(\xi)$ to exist, $f(t)$ is required to be Lebesgue integrable, that is, $\int_{\mathcal{X}} |f(t)| \mathrm{d}t < \infty$.

Observe that $F(\xi)$ is the inner product between the signal $f(t)$ and the Fourier operator $e^{-j2\pi\xi t} = \cos(2\pi\xi t) - j\sin(2\pi\xi t)$, therefore, the complex-valued function $F(\xi)$ contains the frequency content of the even part (cf. odd part) of $f(t)$ in its real part (cf. imaginary part). We also refer to the square absolute value $S(\xi) = |F(\xi)|^2$, which comprises the total frequency content at frequency $\xi$, as the power spectral density (PSD).

Calculating the integral in eq. (1) is far from trivial for general Lebesgue-integrable signals $f(t)$. This has motivated the construction of parametric models for SE that approximate $f(\cdot)$ by analytic expressions that admit closed-form Fourier transform such as sum of sinusoids [8], autoregressive processes [9] and Hermite polynomials. The proposed method will be inspired in this rationale: we will use a stochastic-process model for the signal (rather than a parametric function), to then apply the Fourier transform to such process and finally obtain a stochastic representation of the spectrum. A family of stochastic processes that admit closed-form Fourier transform is presented next.

## 2.3 Gaussian process priors over functions

The Gaussian process (GP [18]) is the infinite-dimensional generalisation of the multivariate normal distribution. Formally, the stochastic process $f(t)$ is a GP if and only if for any finite collection of inputs $\{t_i\}_{i=1}^N$, $N \in \mathbb{N}$, the scalar random variables $\{f(t_i)\}_{i=1}^N$ are jointly Gaussian. A GP $f(t)$ with mean $m$ function and covariance kernel $K$ will be denoted as

$$f(t) \sim \mathcal{GP}(m, K) \tag{2}$$

where we usually assume zero (or constant) mean, and a kernel function $K(t, t')$ denoting the covariance between $f(t)$ and $f(t')$. The behaviour of the GP is encoded in its covariance function, in particular, if the GP $f(t)$ is stationary, we have $K(t, t') = K(t - t')$ and the PSD of $f(t)$ is given by $S(\xi) = \mathcal{F}\{K(t)\}(\xi)$ [24]. The connection between temporal and frequency representations of GPs has aided the design of the GPs to have specific (prior) harmonic content in both parametric [25–29] and non-parametric [22, 23] ways.

GPs are flexible nonparametric models for functions, in particular, for latent signals involved in SE settings. Besides their strength as a generative model, there are two key properties that position GPs as a sound prior within SE: first, as the Fourier transform is a linear operator, the Fourier transform of a GP (if it exists) is also a (complex-valued) GP [30, 31] and, critically, the signal and its spectrum are jointly Gaussian. Second, Gaussian random variables are closed under conditioning and marginalisation, meaning that the exact posterior distribution of the spectrum conditional to a set of partial observations of the signal is also Gaussian. This turns the SE problem into an inference one with two new challenges: to find the requirements for the existence of the spectrum of a GP, and to calculate the statistics of the posterior spectrum given the (temporal) observations.

# 3   A joint generative model for signals and their spectra

The proposed model is presented through the following building blocks: (i) a GP model for the latent signal, (ii) a windowed version of the signal for which the Fourier transform exists, (iii) the closed-form posterior distribution of the windowed-signal spectrum, and (iv) the closed-form posterior power spectral density.

## 3.1   Definition of the local spectrum

We place a stationary GP prior over $f(t) \sim \mathcal{GP}(0, K)$ and model the observations as evaluations of $f(t)$ corrupted by Gaussian noise, denoted by $\mathbf{y} = [y(t_i)]_{i=1}^N$. This GP model follows the implicit stationarity assumption adopted when computing the spectrum via the Fourier transform. However, notice that the draws of a stationary GP are not Lebesgue integrable almost surely (a.s.) and therefore their Fourier transforms do not exist a.s. [32]. We avoid referring to the spectrum of the complete signal and only focus on the spectrum in the neighbourhood of a centre $c$. Then, we can then choose an arbitrarily-wide neighbourhood (as long as it is finite), or consider multiple centres $\{c_i\}_{i=1}^{N_c}$ to form a *bank of filters*. We refer to the spectrum in such neighbourhood as the *local spectrum* and define it through the Fourier transform as

$$F_c(\xi) \triangleq \mathcal{F}\{f_c(t)\} = \mathcal{F}\left\{ f(t - c) e^{-\alpha t^2} \right\} \tag{3}$$

where $f_c(t) = f(t - c) e^{-\alpha t^2}$ is a windowed version of the signal $f(t)$ centred at $c$ with width $1/\sqrt{2\alpha}$. Observe that since $f_c(t)$ decays exponentially for $t \to \pm\infty$, it is in fact Lebesgue integrable:

$$\int_{\mathbb{R}} |f_c(t)| \mathrm{d}t = \int_{\mathbb{R}} |f(t - c) e^{-\alpha t^2}| \mathrm{d}t < \max(|f|) \int_{\mathbb{R}} e^{-\alpha t^2} \mathrm{d}t = \max(|f|) \sqrt{\frac{\pi}{\alpha}} < \infty \quad \text{a.s.} \tag{4}$$

since the $\max(|f|)$ is finite a.s. due to the GP prior. As a consequence, the local spectrum $\mathcal{F}_c\{f(t)\}$ exists and it is finite.

The use of windowed signals is commonplace in SE, either as a consequence of acquisition devices or for algorithmic purposes (as in our case). In fact, windowing allows for a time-frequency representation, meaning that the signal does not need to be stationary but only piece-wise stationary, i.e., different centres $c_i$ might have different spectra. Finally, we clarify that the choice of a square-exponential window $e^{-\alpha t^2}$ obeys to tractability of the statistics calculated in the next section.

A summary of the proposed generative model is shown in eqs. (5)-(8) and a graphical model representation is shown in fig. 1.

$$\text{latent signal:} \quad f(t) \sim \mathcal{GP}(0, K) \tag{5}$$

$$\text{observations:} \quad y(t_i) = f(t_i) + \eta_i, \eta_i \sim \mathcal{N}(0, \sigma_n^2), \forall i = 1 \ldots N, \tag{6}$$

$$\text{windowed signal:} \quad f_c(t) = e^{-\alpha t^2} f(t - c) \tag{7}$$

$$\text{local spectrum:} \quad F_c(\xi) \triangleq \mathcal{F}\{f_c(t)\} = \mathcal{F}\left\{f(t-c)e^{-\alpha t^2}\right\} = \int_{\mathbb{R}} f(t-c)e^{-\alpha t^2}e^{-j2\pi\xi t}\mathrm{d}t \tag{8}$$

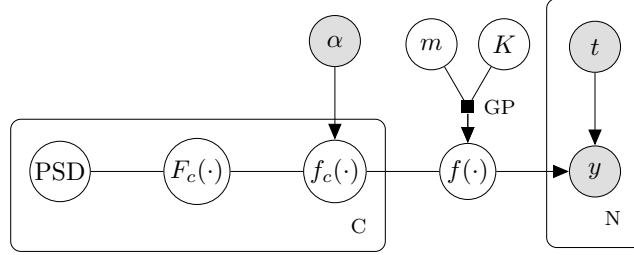

Figure 1: Proposed model for a latent signal $f(t)$, observations $y(t)$, a windowed version $f_c(t)$ and local spectrum $F_c(\xi)$. We have considered $N$ observations and $C$ centres.

## 3.2 The local-spectrum Gaussian process

As a complex-valued linear transformation of $f(t) \sim \mathcal{GP}$, the local spectrum $F_c(\xi)$ is a complex-GP [31, 30] and thus completely determined by its covariance and pseudocovariance [33] given by

$$K_F(\xi, \xi') = \mathbb{E}\left[F_c(\xi)F_c^*(\xi)\right] = \mathbb{E}\left[F_c(\xi)F_c(-\xi')\right] \tag{9}$$

$$P_F(\xi, \xi') = \mathbb{E}\left[F_c(\xi)F_c(\xi')\right] = K_F(\xi, -\xi') \tag{10}$$

where the last identities in each line are due to the fact that the latent function $f(t)$ is real valued.

Recall that we are ultimately interested in the real and imaginary parts of the local spectrum ($\Re F_c(\xi)$ and $\Im F_c(\xi)$ respectively) which are in fact real-valued GPs. However, we will calculate the statistics of the complex-valued $F_c(\xi)$ for notational simplicity, to then calculate the statistics of the real-valued processes $\Re F_c(\xi)$ and $\Im F_c(\xi)$ according to:

$$\text{covariance}(\Re F_c(\xi)) = K_{rr}(\xi, \xi') = \tfrac{1}{2}(K_F(\xi, \xi') + K_F(\xi, -\xi')) \tag{11}$$

$$\text{covariance}(\Im F_c(\xi)) = K_{ii}(\xi, \xi') = \tfrac{1}{2}(K_F(\xi, \xi') - K_F(\xi, -\xi')) \tag{12}$$

$$\text{covariance}(\Re F_c(\xi), \Im F_c(\xi)) = K_{ri}(\xi, \xi') = K_{ir}(\xi, \xi') = 0. \tag{13}$$

The above expressions are due to the identity in eq. (10) and the fact that both $K_F(\xi, \xi')$ and $K_F(\xi, -\xi')$ are real-valued. The relationship between the covariance of a GP and the covariance of the spectrum of such GP is given by the following proposition

**Proposition 1** *The covariance of the local spectrum $F_c(\xi)$ of a stationary signal $f(t) \sim \mathcal{GP}(0, K(t))$ is given by*

$$K_F(\xi, \xi') = \sqrt{\frac{\pi}{2\alpha}}e^{-\frac{\pi^2}{2\alpha}(\xi-\xi')^2}\left(\mathcal{K}(\rho) * \sqrt{\frac{2\pi}{\alpha}}e^{-\frac{2\pi^2}{\alpha}\rho^2}\right)\Bigg|_{\rho=\frac{\xi+\xi'}{2}} \tag{14}$$

*where $\mathcal{K}(\xi) = \mathcal{F}\{K(t)\}(\xi) = \int_{\mathbb{R}} K(t)e^{-j2\pi\xi t}dt$ is the Fourier transform of the kernel $K$. Equivalently, as pointed out in eq. (10), the pseudocovariance is given by replacing the above expression in $P_F(\xi, \xi') = K_F(-\xi, \xi')$.*

See the proof in Section 1.1 of the supplementary material. Notice that the covariance of the local spectrum $K_F$ is a sequence of linear transformations of the covariance of the signal $K$ according to: (i) the Fourier transform due to the domain change, (ii) convolution with $e^{-\frac{2\pi^2}{\alpha}\rho^2}$ due to windowing effect, and (iii) a smoothness factor $e^{-\frac{\pi^2}{2\alpha}(\xi-\xi')^2}$ that depends on the window width; this means that

for wider windows the values of the local spectrum at different frequencies become independent. Critically, observe that each of the Gaussian functions in eq. (14) are divided by their normalising constants, therefore the norm of $K_F$ is equal to the norm $\mathcal{K}$, which is in turn equal to the norm of the covariance of the signal $K$ due to the unitary property of the Fourier transform.

With an illustrative purpose, we evaluate $K_F$ for the $Q$-component spectral mixture (SM) kernel [26]

$$K_{\text{SM}}(\tau) = \sum_{q=1}^{Q} \sigma_q^2 \exp\left(-\gamma_q \tau^2\right) \cos(2\pi \theta_q^\top \tau) \tag{15}$$

the Fourier transform of which is known explicitly and given by

$$\mathcal{K}_{\text{SM}}(\xi) = \sum_{q=1}^{Q} \sigma_q^2 \sqrt{\frac{\pi}{\gamma_q}} \left( \frac{e^{-\frac{\pi^2}{\gamma_q}(\xi-\theta_q)^2} + e^{-\frac{\pi^2}{\gamma_q}(\xi+\theta_q)^2}}{2} \right) = \sum_{q=1}^{Q} \sum_{\theta=\pm\theta_q} \frac{\sigma_q^2}{2} \sqrt{\frac{\pi}{\gamma_q}} e^{-\frac{\pi^2}{\gamma_q}(\xi-\theta_q)^2}. \tag{16}$$

For this SM kernel, the covariance kernel of the local-spectrum process is (see supp. mat., §1.2)

$$K_F(\xi,\xi') = \sum_{q=1}^{Q} \sum_{\theta=\pm\theta_q} \frac{\sigma_q^2 \pi}{2\sqrt{\alpha(\alpha+2\gamma_q)}} e^{-\frac{\pi^2}{2\alpha}(\xi-\xi')^2} e^{-\frac{2\pi^2}{\alpha+2\gamma_q}\left(\frac{\xi+\xi'}{2}-\theta_q\right)^2}. \tag{17}$$

With the explicit expression of $K_F$ in eq. (17) and the relationships in eqs. (9)-(13), we can compute the statistics of the real and imaginary parts of the local spectrum and sample from it. Fig. 2 shows these covariances and 3 sample paths revealing the odd and even properties of the covariances.

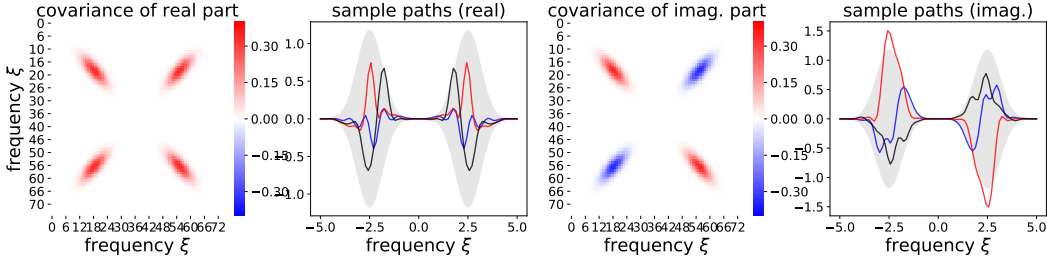

Figure 2: Covariance and sample paths of the local-spectrum of a SM signal with $Q = 1, \sigma_q = 1, \gamma_q = 5e-3, \theta_q = 2.5, \alpha = 5e-5$. Real (cf. imaginary) part shown in the left (cf. right) half.

### 3.3 Joint samples and the conditional density $p(F_c(\xi)|\mathbf{y})$

Although the joint distribution over the signal $f(t)$ and its local spectrum $\mathcal{F}_c(\xi)$ is Gaussian, sampling directly from this joint distribution is problematic due to the deterministic relationship between the (complete and noiseless) signal $f$ and its local spectrum. We thus proceed hierarchically: we first sample $\mathbf{y} \sim \mathcal{GP}(\mathbf{t}; 0, K), \mathbf{y} \in \mathbb{R}^N$, and then $\mathcal{F}_c(\xi) \sim p(\mathcal{F}_c|\mathbf{y})$, where the posterior is normally-distributed with mean and covariance given respectively by

$$\mathbb{E}\left[F_c(\xi)|\mathbf{y}\right] = K_{\mathbf{y}F_c}^\top(\mathbf{t},\xi)K(\mathbf{t},\mathbf{t})^{-1}\mathbf{y} \tag{18}$$

$$\mathbb{E}\left[F_c^*(\xi)F_c(\xi')|\mathbf{y}\right] = K_F(\xi,\xi') - K_{\mathbf{y}F_c}^\top(\mathbf{t},\xi)K(\mathbf{t},\mathbf{t})^{-1}K_{\mathbf{y}F_c}(\mathbf{t},\xi) \tag{19}$$

where $K_{\mathbf{y}F_c}(\mathbf{t},\xi)$ is presented in the next proposition.

**Proposition 2** *The covariance $K_{\mathbf{y}F_c}(\mathbf{t},\xi)$ between the observations $\mathbf{y}$ at times $\mathbf{t}$ coming from a stationary signal $f(t) \sim \mathcal{GP}(0, K)$ and its local spectrum at frequency $\xi$ is given by*

$$K_{yF_c}(t,\xi) = \mathbb{E}\left[y_c^*(t)F_c(\xi)\right] = \mathcal{K}(\xi)e^{-j2\pi\xi t} * \sqrt{\frac{\pi}{\alpha}}e^{-\frac{\pi^2\xi^2}{\alpha}} \tag{20}$$

*where $\mathcal{K}(\xi) = \mathcal{F}\{K(t)\}(\xi) = \int_{\mathbb{R}} K(t)e^{-j2\pi\xi t}dt$ is the Fourier transform of the kernel $K$.*

See the proof in Section 1.3 of the supplementary material. Notice that the convolution against $e^{-\frac{\pi^2\xi^2}{\alpha}}$ is also due to the windowing effect and that the norms of $K_{yF_c}$ and $K$ are equal.

For the SM kernel, shown in eq.(15), $K_{yF_c}$ becomes (details in supp. mat., §1.4)

$$K_{y\mathcal{F}} = \sum_{q=1}^{Q} \sum_{\theta=\pm\theta_q} \frac{\sigma_q^2}{2\sqrt{\pi(\tilde{\alpha}+\tilde{\gamma}_q)}} \exp\left(-\frac{(\xi-\theta_q)^2}{\tilde{\alpha}+\tilde{\gamma}_q}\right) \exp\left(-\frac{\pi^2 t^2}{L_q}\right) \exp\left(-j\frac{2\pi t}{L_q}\left(\frac{\theta_q}{\tilde{\gamma}_q}+\frac{\xi}{\tilde{\alpha}}\right)\right)$$

where $\tilde{\alpha} = \alpha/\pi^2$, $\tilde{\gamma}_q = \gamma_q/\pi^2$ and $L_q = (\tilde{\alpha}^{-1} + \tilde{\gamma}_q^{-1})^{-1}$. Fig. 3 shows this covariance together with joint samples of the signal and its spectrum (colour-coded).

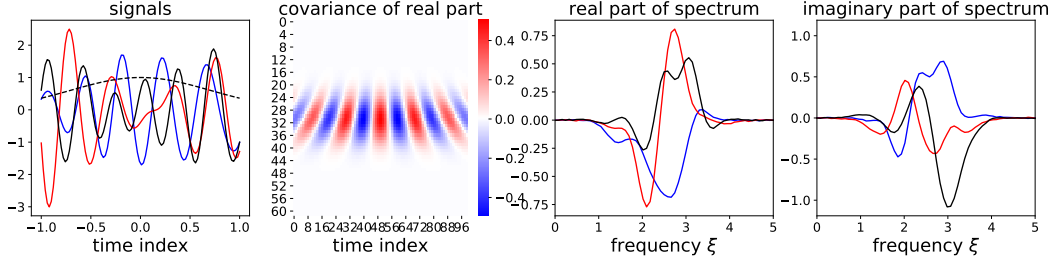

Figure 3: Hierarchical sampling. From left to right: Signal samples (solid) and window (dashed), covariance $K_{y\mathcal{F}}$ for the SM, real-part local-spectrum samples, imaginary-part local-spectrum. Parameters were $Q = 1, \sigma_q = 1, \gamma_q = 2, \theta_q = 2.5, \alpha = 1$. Notice how $K_{yF_c}(t,\xi)$ vanishes as the frequency $\xi$ departs from $\theta_q$.

We conclude this section with the following result.

**Proposition 3** *The power spectral density of a stationary signal $f(t) \sim \mathcal{GP}(0, K)$, conditional to a set of observations* $\mathbf{y}$, *is a $\chi^2$-distributed stochastic process and its mean is known in closed form.*

This result follows from the fact the (posterior) real and imaginary parts of the spectrum are independent Gaussian process with explicit mean and covariance. This is a critical contribution of the proposed model, where the search for periodicities can be performed by optimising a closed-form expression which has a linear evaluation cost.

## 4  Spectral estimation as Bayesian inference

Henceforth, the proposed method for Bayesian nonparametric spectral estimation will be referred to as BNSE. This section analyses BNSE in terms of interpretability, implementation, and connection with other methods.

### 4.1  Training and computational cost

BNSE can be interpreted as fitting a continuous-input interpolation to the observations, computing the Fourier transform of the interpolation and finally average over all the possibly infinitely-many interpolations. Consequently, as our interpolation is a GP, both the Fourier transform and the infinite average can be performed analytically. Within BNSE, finding the appropriate interpolation family boils down to selecting the model hyperparameters, where the GP prior protects the model from overfitting [18]. In this regard, the proposed BNSE can readily rely upon state-of-the-art training procedures for GPs and benefit from sparse approximations for computationally-efficient training. Finally, as the hyperparameters of the posterior spectrum are given by those of the GP in the time domain, computing the posterior local spectrum poses no additional computational complexity.

### 4.2  Model consistency and interpretation

The problem of global (rather than local) SE can be addressed by choosing an arbitrarily-wide window. However, as pointed out in Section 3.1 recall that the local-spectrum process is not defined

for $\alpha \to 0$, since it turns into the sum of infinitely-many Gaussian RVs; in fact, note from eq. (14) that $\lim_{\alpha \to 0} K_F(\xi, \xi') = \infty$. Despite the lack of convergence for the posterior law of the spectrum when $\alpha \to 0$, let us only consider the point estimate as the posterior mean defined from eqs. (18) and (20) as

$$\mathbb{E}\left[\mathcal{F}_c(\xi)|\mathbf{y}\right] = \left(\mathcal{K}(\xi)e^{-j2\pi\xi t} * \sqrt{\frac{\pi}{\alpha}}e^{-\frac{\pi^2\xi^2}{\alpha}}\right)K(\mathbf{t}, \mathbf{t})^{-1}\mathbf{y} \tag{21}$$

Observe that we can indeed apply the limit $\alpha \to 0$ above, where the second argument of the convolution converges to a (unit-norm) Dirac delta function. Additionally, let us consider an uninformative prior over the latent signal by choosing $K(\mathbf{t}, \mathbf{t}) = \mathbf{I}$, which implies $\mathcal{K}(\xi) = 1$. Under these conditions (infinitely-wide window and uninformative prior for temporal structure in the signal) the point estimate of the proposed model becomes the discrete-time Fourier transform.

$$\lim_{\alpha \to 0}\mathbb{E}\left[\mathcal{F}_c(\xi)|\mathbf{y}\right] = e^{-j2\pi\xi\mathbf{t}}\mathbf{y} = \sum_{i=1}^{N}e^{-j2\pi\xi t_i}y(t_i). \tag{22}$$

This reveals the consistency of the model and offers a clear interpretation of the functional form in eq. (21): the posterior mean of the local spectrum is a linear transformation of a whitened version of the observations that depends on the width of the window and the prior belief over frequencies.

### 4.3 Approximations for non-exponential covariances

Though Sec. 3 provides explicit expressions of the posterior local-spectrum statistics for the spectral mixture kernel [26], the proposed method is independent of the stationary kernel considered. For general kernels with known Fourier transform but for which the convolutions in eqs. (14) and (20) are intractable such as the Sinc, Laplace and Matérn kernels [34], we consider the following approximation for $\alpha$ sufficiently small

$$K_F(\xi, \xi') = \frac{\pi}{\alpha}e^{-\frac{\pi^2}{2\alpha}(\xi-\xi')^2}\left(\mathcal{K}(\rho) * e^{-\frac{2\pi^2}{\alpha}\rho^2}\right)\Bigg|_{\rho=\frac{\xi+\xi'}{2}} \approx \sqrt{\frac{\pi}{2\alpha}}e^{-\frac{\pi^2}{2\alpha}(\xi-\xi')^2}\mathcal{K}\left(\frac{\xi+\xi'}{2}\right) \tag{23}$$

$$K_{y_c\mathcal{F}_c}(t, \xi) = \mathcal{K}(\xi)e^{-j2\pi\xi t} * \sqrt{\frac{\pi}{\alpha}}e^{-\frac{\pi^2\xi^2}{\alpha}} \approx \mathcal{K}(\xi)e^{-j2\pi\xi t} \tag{24}$$

where we approximated the second argument in both convolutions as a Dirac delta as in Sec. 4.2. We did not approximate the term $\sqrt{\frac{\pi}{2\alpha}}e^{-\frac{\pi^2}{2\alpha}(\xi-\xi')^2}$ in eq. (23) since placing a Dirac delta outside a convolution will result on a degenerate covariance. We emphasise that this is an approximation for numerical computation and not applying the limit $\alpha \to 0$, in which case BNSE does not converge.

### 4.4 Proposed model as the limit of the Lomb-Scargle method

The Lomb-Scargle method (LS) [8] is the *de facto* approach for estimating the spectrum of nonuniformly-sampled data. LS proceeds by fitting a set of sinusoids via least squares to the observations and then reporting the estimated spectrum as the weights of the sinusoids. The proposed BNSE method is closely related to the LS method with clear differences: (i) we assume a probabilistic model (the GP) which allows for the spectrum to be stochastic, (ii) we assume a nonparametric model which expressiveness increases with the amount of data, (iii) BNSE is trained once and results in a functional form for $\mathcal{F}_c(\xi)$, whereas LS needs to be retrained should new frequencies be considered, (iv) the functional form $\mathcal{F}_c(\xi)$ allows for finding periodicities via optimisation, while LS can only do so through exhaustive search and retraining in each step. In Section 2 of the supplementary material, we show that the proposed BNSE model is the limit of the LS method when an infinite number of components is considered with a Gaussian prior over the weights.

## 5 Simulations

This experimental section contains three parts focusing respectively on: (i) consistency of BNSE in the classical sum-of-sinusoids setting, (ii) robustness of BNSE to overfit and ability to handle non-uniformly sampled noisy observations (heart-rate signal), and (iii) exploiting the functional form of the PSD estimate of BNSE to find periodicities (astronomical signal).

## 5.1 Identifying line spectra

Signals composed by a sum of sinusoids have spectra given by Dirac delta functions (or vertical lines) referred to as *line spectra*. We compared BNSE against classic line spectra models such as MUSIC [7], Lomb-Scargle [8] and the Periodogram [10]. We considered 240 evenly-sampled observations of the signal $f(t) = 10\cos(2\pi 0.5t) - 5\sin(2\pi 1.0t)$ in the domain $\mathbf{t} \in [-10, 10]$ corrupted by zero-mean unit-variance Gaussian noise. The window parameter was set to $\alpha = 1/(2 \cdot 50^2)$ for an observation neighbourhood much wider than the support of the observations, and we chose an SM kernel with rather permissive hyperparameters: a rate $\gamma = 1/(2 \cdot 0.05^2)$ and $\theta = 0$ for a prior over frequencies virtually uninformative. Fig. 4 shows the real and imaginary parts of the posterior local spectrum and the sample PSD against LS, MUSIC, and the Periodogram. Notice how BNSE recovered the spectrum with tight error bars and appropriate relative magnitudes. Additionally, from the PSD estimates notice how both BNSE and LS coincided with the periodogram and MUSIC at the peaks of the PSD. Finally, observe that in line with the structural similarities between BNSE and LS, they both exhibit the same lobewidths and that LS falls within the errorbars of BNSE.

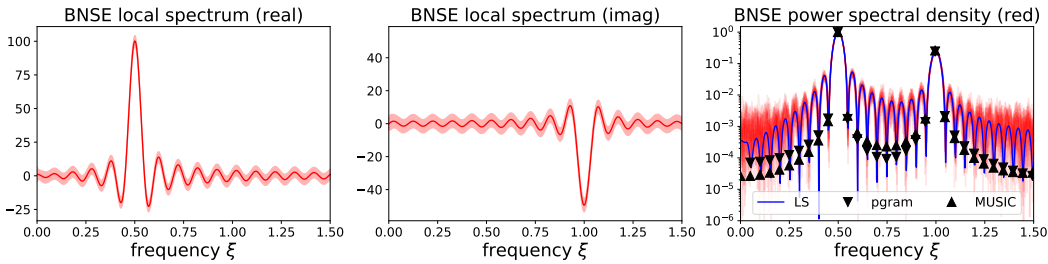

Figure 4: Line spectrum estimates: BNSE is shown in red and its PSD is computed by first sampling form the real and imaginary parts of the posterior spectrum and then adding the square samples (LS: Lomb-Scargle and pgram: periodogram).

s

## 5.2 Discriminating between heart-rate signals

We next considered two heart-rate signals from `http://ecg.mit.edu/time-series/`. The first one is known to have frequency components at the respiration rate of the subject, whereas the second one exhibits low-frequency energy which may be attributed to congestive heart failure [35]. To show that the proposed method does not overfit to the spectrum of the training data, we used the first signal to train BNSE and then used BNSE to analyse the posterior PSD of the second signal. To make the experiment more realistic, we only used an unevenly-sampled 10% of the data from the second (test) signal and considered the LS method with the entire (noiseless) signal as ground truth. Fig. 5 shows the PSDs for both signals and methods. Observe that in both cases, BNSE's posterior PSD distribution includes the ground truth (LS), even for the previously-unseen test signal. Crucially, this reveals that BNSE can be used for SE beyond the training data to find critical harmonic features from noisy and limited observations.

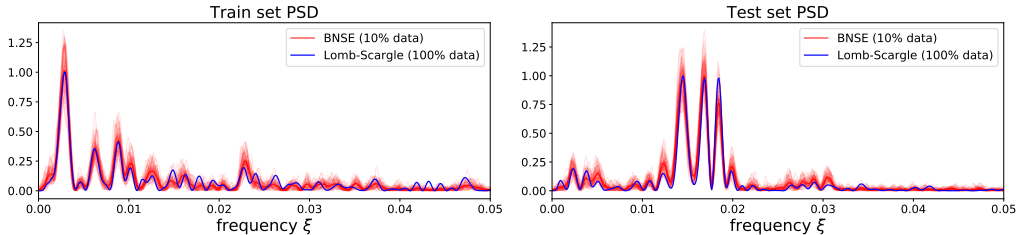

Figure 5: PSD estimates for heart-rate time series. Notice how BNSE recovered the spectral content of the test signal from only a few noisy measurements

### 5.3 Finding periodicities via efficient optimisation

Lastly, we considered the sunspots dataset, an astronomical time series that is known to have a period of approximately 11 years, corresponding to a fundamental frequency of $1/11 \approx 0.089$. Finding this period is challenging due to the nonstationarity of the signal. We implemented BNSE, Lomb-Scargle and a GP with spectral mixture kernel [26] to find the fundamental frequency of the series. Satisfactory training for Lomb-Scargle and the SM kernel was not possible via gradient-based maximum likelihood (we used GPflow [36]), even starting from the neighbourhood of the true frequency (0.089) or using minibatches. Our conjecture is that this is due to the fact that the sunspots series is neither strictly periodic nor Gaussian. We implemented BNSE with a lengthscale equal to one and $\theta = 0$ for a broad prior over frequencies, and $\alpha = 10^{-3}$ for a wide observation neighbourhood. Finally, the posterior mean of the PSD reported by BNSE was maximised using the derivative-free Powell method [37] due to its non-convexity. Notice that optimising the PSD of BNSE with Powell has a computational cost that is linear in the number observations and dimensions, whereas maximising SM via maximum likelihood has a cubic cost in the observations. Fig. 6 shows the estimates of the PSD for BNSE and LS (recall that SM was not able to train) and their maxima, observe how the global maximum of the PSD estimated by BNSE is the true period $\approx 0.089$.

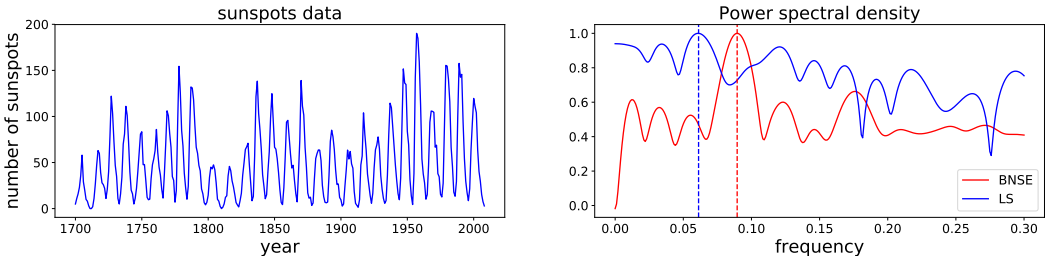

Figure 6: Finding periodicities via optimisation. Left: sunspots data. Right: PSDs estimates reported by BNSE (red) and LS (blue) with corresponding maxima in vertical dashed lines. The correct fundamental frequency of the series is approximately $1/11 \approx 0.089$.

## 6 Discussion

We have proposed a nonparametric model for spectral estimation (SE), termed BNSE, and have shown that it admits exact Bayesian inference. BNSE builds on a Gaussian process (GP) prior over signals and its relationship to existing methods in the SE and GP literature has been illuminated from both theoretical and experimental perspectives. To the best of our knowledge, BNSE is the first nonparametric approach for SE where the representation of uncertainty related to missing and noisy observations is inherent (due to its Bayesian nature). Another unique advantage of BNSE is a nonparametric functional form for the posterior power spectral density (PSD), meaning that periodicities can be found through linear-cost optimisation of the PSD rather than by exhaustive search or expensive non-convex optimisation routines. We have shown illustrative examples and results using time series and exponential kernels, however, the proposed BNSE is readily available to take full advantage of GP theory to consider arbitrary kernels in multi-input, multi-output, nonstationary and even non-Gaussian applications. The promising theoretical results also open new avenues in modern SE, this may include novel interpretations of Nyquist frequency, band-pass filtering and time-frequency analysis.

**Acknowledgments**

This work was funded by the projects Conicyt-PIA #AFB170001 Center for Mathematical Modeling and Fondecyt-Iniciación #11171165.

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
