[Supplementary Material]

# Bayesian Nonparametric Spectral Estimation: Supplementary Material

**Felipe Tobar**
Universidad de Chile
`ftobar@dim.uchile.cl`

## 1 Calculations required for the covariances of the local spectrum GP

### 1.1 Prior covariance of local spectrum: general case

The covariance of values of the local spectrum $\mathcal{F}_c\left\{f(t)\right\}$ and $\mathcal{F}_c\left\{f(t')\right\}$ is given by:

$$
\begin{aligned}
K_F(\xi,\xi') &= \mathbb{E}\left[\mathcal{F}_c\left\{f(t)\right\}^*(\xi)\mathcal{F}_c\left\{f(t')\right\}(\xi')\right] && \text{def. } K_F \\
&= \mathbb{E}\left[\mathcal{F}\left\{f(t-c)e^{-\alpha t^2}\right\}^*(\xi)\mathcal{F}\left\{f(t'-c)e^{-\alpha t'^2}\right\}(\xi')\right] && \text{def. } \mathcal{F}_c \\
&= \mathbb{E}\left[\int_{\mathbb{R}}f(t-c)e^{-\alpha t^2}e^{j2\pi\xi t}\mathrm{d}t\int_{\mathbb{R}}f(t'-c)e^{-\alpha t'^2}e^{-j2\pi\xi' t'}\mathrm{d}t'\right] && \text{def. Fourier transform } \mathcal{F} \\
&= \int_{\mathbb{R}^2}e^{j2\pi\xi t}e^{-\alpha t^2}\mathbb{E}\left[f(t-c)f(t'-c)\right]e^{-\alpha t'^2}e^{-j2\pi\xi' t'}\mathrm{d}t\mathrm{d}t' && \text{switch integrals and } \mathbb{E}\left[\cdot\right]\text{ (Fubini)} \\
&= \int_{\mathbb{R}^2}e^{j2\pi\xi t}e^{-\alpha t^2}K(t-t')e^{-\alpha t'^2}e^{-j2\pi\xi' t'}\mathrm{d}t\mathrm{d}t' && \text{def. } K(t) \\
&= \mathcal{F}\left\{e^{-\alpha t^2}K(t-t')e^{-\alpha t'^2}\right\}(-\xi,\xi') && \text{def. Fourier transform } \mathcal{F} \\
&= \mathcal{F}\left\{K(t-t')\right\}(-\xi,\xi')*\mathcal{F}\left\{e^{-\alpha t^2-\alpha t'^2}\right\}(-\xi,\xi') && \text{convolution thm.} \\
&= \mathcal{F}\left\{K(\tau)\right\}(-\xi)\delta(\xi-\xi')*\frac{\pi}{\alpha}e^{-\frac{\pi^2}{\alpha}(\xi^2+\xi'^2)} && \text{rearrange} \\
&= \frac{\pi}{\alpha}\int_{\mathbb{R}^2}\mathcal{K}(\lambda)\delta(\lambda-\lambda')e^{-\frac{\pi^2}{\alpha}((\xi-\lambda)^2+(\xi'-\lambda')^2)}\mathrm{d}\lambda\mathrm{d}\lambda' && \text{def. convolution} \\
&= \frac{\pi}{\alpha}\int_{\mathbb{R}}\mathcal{K}(\lambda)e^{-\frac{\pi^2}{\alpha}((\xi-\lambda)^2+(\xi'-\lambda)^2)}\mathrm{d}\lambda && \text{integrate wrt } \lambda' \\
&= \frac{\pi}{\alpha}\int_{\mathbb{R}}\mathcal{K}(\lambda)e^{-\frac{\pi^2}{\alpha}\left(2\left(\lambda-\frac{\xi+\xi'}{2}\right)^2+\frac{1}{2}(\xi-\xi')^2\right)}\mathrm{d}\lambda && \text{rearrange} \\
&= \frac{\pi}{\alpha}e^{-\frac{\pi^2}{2\alpha}(\xi-\xi')^2}\int_{\mathbb{R}}\mathcal{K}(\lambda)e^{-\frac{2\pi^2}{\alpha}\left(\frac{\xi+\xi'}{2}-\lambda\right)^2}\mathrm{d}\lambda && \text{rearrange} \\
&= \frac{\pi}{\alpha}e^{-\frac{\pi^2}{2\alpha}(\xi-\xi')^2}\left.\left(\mathcal{K}(\rho)*e^{-\frac{2\pi^2}{\alpha}\rho^2}\right)\right|_{\rho=\frac{\xi+\xi'}{2}} && \text{def. convolution}
\end{aligned}
$$

where $\mathcal{K}(\xi)=\mathcal{F}\left\{K(t)\right\}(\xi)=\int_{\mathbb{R}}K(t)e^{-j2\pi\xi t}\mathrm{d}t$ is the Fourier transform of the kernel $K$.

## 1.2 Prior covariance of local spectrum: spectral mixture case

Replacing the spectral mixture kernel $K_{\text{SM}}(\tau) = \sum_{q=1}^{Q} \sigma_q^2 \exp\left(-\gamma_q \tau^2\right) \cos(2\pi \theta_q^\top \tau)$ in the above expression, we have

$$K_F(\xi, \xi') = \sum_{q=1}^{Q} \sum_{\theta=\pm\theta_q} \frac{\pi}{\alpha} e^{-\frac{\pi^2}{2\alpha}(\xi-\xi')^2} \left( \frac{\sigma_q^2}{2} \sqrt{\frac{\pi}{\gamma_q}} e^{-\frac{\pi^2}{\gamma_q}(\rho-\theta)^2} * e^{-\frac{2\pi^2}{\alpha}\rho^2} \right) \Bigg|_{\rho=\frac{\xi+\xi'}{2}} \tag{1}$$

$$= \sum_{q=1}^{Q} \sum_{\theta=\pm\theta_q} \frac{\sigma_q^2 \pi^{3/2}}{2\alpha \sqrt{\gamma_q}} e^{-\frac{\pi^2}{2\alpha}(\xi-\xi')^2} \frac{\sqrt{\pi}}{\sqrt{\pi^2/\gamma_q + 2\pi^2/\alpha}} e^{-\frac{(\rho-\theta)^2 \pi^2 \pi^2 2/(\alpha\gamma_q)}{\pi^2/\gamma_q + 2\pi^2/\alpha}} \Bigg|_{\rho=\frac{\xi+\xi'}{2}} \tag{2}$$

$$= \sum_{q=1}^{Q} \sum_{\theta=\pm\theta_q} \frac{\sigma_q^2 \pi}{2\sqrt{\alpha(\alpha+2\gamma_q)}} e^{-\frac{\pi^2}{2\alpha}(\xi-\xi')^2} e^{-\frac{2\pi^2(\rho-\theta)^2}{\alpha+2\gamma_q}} \Bigg|_{\rho=\frac{\xi+\xi'}{2}} \tag{3}$$

$$= \sum_{q=1}^{Q} \sum_{\theta=\pm\theta_q} \frac{\sigma_q^2 \pi}{2\sqrt{\alpha(\alpha+2\gamma_q)}} e^{-\frac{\pi^2}{2\alpha}(\xi-\xi')^2} e^{-\frac{2\pi^2}{\alpha+2\gamma_q}\left(\frac{\xi+\xi'}{2}-\theta\right)^2} \tag{4}$$

## 1.3 Covariance between the signal $y$ and the local spectrum $\mathcal{F}_c(\xi)$: general case

$$K_{y_c \mathcal{F}_c}(t, \xi) = \mathbb{E}\left[y_c^*(t) \mathcal{F}_c(\xi)\right] \tag{5}$$

$$= \mathbb{E}\left[(f(t-c) + \epsilon) \int_{\mathbb{R}} f(\tau-c) e^{-\alpha\tau^2} e^{-j2\pi\xi\tau} d\tau\right] \tag{6}$$

$$= \int_{\mathbb{R}} \mathbb{E}\left[f(t-c) f(\tau-c)\right] e^{-\alpha\tau^2} e^{-j2\pi\xi\tau} d\tau \tag{7}$$

$$= \int_{\mathbb{R}} K(\tau-t) e^{-\alpha\tau^2} e^{-j2\pi\xi\tau} d\tau \tag{8}$$

$$= \mathcal{F}\left\{K(\tau-t) e^{-\alpha\tau^2}\right\}(\xi) \tag{9}$$

$$= \mathcal{F}\left\{K(\tau-t)\right\}(\xi) * \mathcal{F}\left\{e^{-\alpha\tau^2}\right\}(\xi) \tag{10}$$

$$= \mathcal{K}(\xi) e^{-j2\pi\xi t} * \sqrt{\frac{\pi}{\alpha}} e^{-\frac{\pi^2 \xi^2}{\alpha}} \tag{11}$$

### 1.4 Covariance between the signal $y$ and the local spectrum $\mathcal{F}_c(\xi)$: Gaussian mixture case

Let us first compute the convolution term $\text{CT}_q = \left( e^{-\frac{\pi^2}{\gamma_q}(\xi-\theta)^2} e^{-j2\pi\xi t} \right) * e^{-\frac{\pi^2\xi^2}{\alpha}}$:

$$
\begin{aligned}
\text{CT} &= \int_{\mathbb{R}} \exp\left( -\frac{\pi^2}{\gamma_q}(\lambda-\theta)^2 - j2\pi\lambda t - \frac{\pi^2(\xi-\lambda)^2}{\alpha} \right) d\lambda \\
&= \int_{\mathbb{R}} \exp\left( -\frac{\pi^2}{\gamma_q}(\lambda^2 - 2\lambda\theta + \theta^2) - j2\pi\lambda t - \frac{\pi^2}{\alpha}(\xi^2 - 2\xi\lambda + \lambda^2) \right) d\lambda \\
&= \int_{\mathbb{R}} \exp\left( -\lambda^2\left(\frac{\pi^2}{\gamma_q} + \frac{\pi^2}{\alpha}\right) + 2\lambda\left(\frac{\pi^2\theta}{\gamma_q} + \frac{\pi^2\xi}{\alpha} - j\pi t\right) - \frac{\pi^2\theta^2}{\gamma_q} - \frac{\pi^2\xi^2}{\alpha} \right) d\lambda \\
&= \int_{\mathbb{R}} \exp\left( -\lambda^2 \underbrace{\left(\frac{1}{\tilde{\gamma}_q} + \frac{1}{\tilde{\alpha}}\right)}_{L_q} + 2\lambda\left(\frac{\theta}{\tilde{\gamma}_q} + \frac{\xi}{\tilde{\alpha}} - j\pi t\right) - \frac{\theta^2}{\tilde{\gamma}_q} - \frac{\xi^2}{\tilde{\alpha}} \right) d\lambda \qquad\qquad \tilde{\alpha} = \frac{\alpha}{\pi^2}, \tilde{\gamma} = \frac{\gamma}{\pi^2} \\
&= \sqrt{\frac{\pi}{L_q}} \exp\left( \frac{1}{L_q}\left(\frac{\theta}{\tilde{\gamma}_q} + \frac{\xi}{\tilde{\alpha}} - j\pi t\right)^2 - \frac{\theta^2}{\tilde{\gamma}_q} - \frac{\xi^2}{\tilde{\alpha}} \right) \\
&= \sqrt{\frac{\pi}{L_q}} \exp\left( \xi^2 \underbrace{\left(\frac{1}{L_q\tilde{\alpha}^2} - \frac{1}{\tilde{\alpha}^2}\right)}_{-(\tilde{\alpha}+\tilde{\gamma})^{-1}} + 2\xi\theta \underbrace{\left(\frac{1}{L_q\tilde{\alpha}\tilde{\gamma}}\right)}_{-(\tilde{\alpha}+\tilde{\gamma})^{-1}} + \theta^2 \underbrace{\left(\frac{1}{L_q\tilde{\gamma}_q^2} - \frac{1}{\tilde{\gamma}_q^2}\right)}_{-(\tilde{\alpha}+\tilde{\gamma})^{-1}} - j\frac{2\pi t}{L_q}\left(\frac{\theta}{\tilde{\gamma}_q} + \frac{\xi}{\tilde{\alpha}}\right) - \frac{\pi^2 t^2}{L_q} \right) \\
&= \sqrt{\frac{\pi}{L_q}} \exp\left( -\frac{(\xi-\theta)^2}{\tilde{\alpha}+\tilde{\gamma}} \right) \exp\left( -\frac{\pi^2 t^2}{L_q} \right) \exp\left( -j\frac{2\pi t}{L_q}\left(\frac{\theta}{\tilde{\gamma}_q} + \frac{\xi}{\tilde{\alpha}}\right) \right)
\end{aligned}
$$

now calculate $K_{y\mathcal{F}}$ and replace for the above term

$$
\begin{aligned}
K_{y\mathcal{F}} &= \mathcal{K}(\xi)e^{-j2\pi\xi t} * \sqrt{\frac{\pi}{\alpha}}e^{-\frac{\pi^2\xi^2}{\alpha}} \\
&= \sum_{q=1}^{Q}\sum_{\theta=\pm\theta_q} \frac{\sigma_q^2}{2}\sqrt{\frac{\pi}{\gamma_q}}\sqrt{\frac{\pi}{\alpha}} \underbrace{e^{-\frac{\pi^2}{\gamma_q}(\xi-\theta)^2} e^{-j2\pi\xi t} * e^{-\frac{\pi^2\xi^2}{\alpha}}}_{\text{CT}_q} \\
&= \sum_{q=1}^{Q}\sum_{\theta=\pm\theta_q} \frac{\sigma_q^2}{2\sqrt{\pi(\tilde{\alpha}+\tilde{\gamma}_q)}} \exp\left( -\frac{(\xi-\theta)^2}{\tilde{\alpha}+\tilde{\gamma}} \right) \exp\left( -\frac{\pi^2 t^2}{L_q} \right) \exp\left( -j\frac{2\pi t}{L_q}\left(\frac{\theta}{\tilde{\gamma}_q} + \frac{\xi}{\tilde{\alpha}}\right) \right)
\end{aligned}
$$

where $\tilde{\alpha} = \alpha/\pi^2, \tilde{\gamma} = \gamma/\pi^2$ and $L = (\tilde{\alpha}^{-1} + \tilde{\gamma}^{-1})^{-1}$

## 2 Proposed model as the limit of the Lomb-Scargle method

Let us consider the model assumed by Lomb-Scargle (LS)

$$
f_S(t) = \sum_{i=1}^{S} a_i \cos(\omega_i t) + b_i \sin(\omega_i t) \tag{12}
$$

where $\{\omega_i\}_{i=1}^{S}$ are fixed frequencies and the weights $\mathbf{a} = [a_i]_{i=1}^{S}$ and $\mathbf{b} = [b_i]_{i=1}^{S}$ are the free parameters. We have used angular-frequency notation according to the original formulation of the LS method, this can be converted to natural frequencies used in the rest of the paper by $\omega = 2\pi\xi$.

We convert the expression in eq.(12) into a probabilistic model by equipping it with prior distribution over the weights, this priori is chosen so that $\mathbf{a}$ and $\mathbf{b}$ are independent from one another and are both normally distributed with zero mean and variance $\Sigma = [\Sigma_{i,j}]_{i,j=1}^{S}$, that is

$$
p(\mathbf{a}, \mathbf{b}) = p(\mathbf{a})p(\mathbf{b}) = \mathcal{N}(\mathbf{a}; 0, \Sigma)\mathcal{N}(\mathbf{b}; 0, \Sigma) \tag{13}
$$

Accordingly, $f_S$ is a Gaussian process (GP), as it is a sum of basis functions with Gaussian weights. The mean of $f_S$ is zero and its covariance is given by

$$K(t,t') = \mathbb{E}\left[f_S(t)f_S(t')\right] = \sum_{ij=1}^{S} \Sigma_{i,j} \cos(\omega_i t - \omega_j t') \tag{14}$$

This is now a GP generative model for the latent function with a nonstationary covariance function (not a function of $t - t'$) arising by the choice of a finite number of frequencies $\omega \in \mathbb{R}$. Considering and infinite number of frequencies and replacing $\omega_i = \omega$ and $\omega_i = \omega'$ for notational consistency with the infinite-dimensional case, we have

$$K(t,t') = \int_{\mathbb{R}^2} K(\omega,\omega') \cos(\omega_i t - \omega_j t') d\omega d\omega'. \tag{15}$$

To calculate the above expression explicitly, we choose covariance[1] as

$$K(\omega,\omega') = \sigma^2 \delta_{\omega-\omega'} e^{-\gamma(\omega-\theta)^2} e^{-\gamma(\omega'-\theta)^2} \tag{16}$$

meaning that nonzero weights are only possible for frequencies sufficiently close to $\theta$ and that the weights for different frequencies are uncorrelated. Replacing $K(\omega,\omega')$ into eq. (15), we obtain

$$K(t,t') = \frac{\pi}{2\sqrt{2\gamma}} \exp(2\gamma\theta^2) \exp\left(-\frac{(t-t')^2}{8\gamma}\right) \cos(\theta(t-t')) \tag{17}$$

that is, the spectral mixture kernel considered above.

Therefore, we have shown that when the model assumed by the Lomb-Scargle model is considered with an infinite number of components, and a Gaussian prior over the weights as defined in eq. (16), it converges to the generative model used in the proposed BNSE approach.

## Footnotes

[1]The Dirac delta comes from $\lim_{\alpha\to\infty} \sqrt{\frac{\alpha}{\pi}} e^{-\alpha(\omega-\omega')^2}$