[Reviews · NeurIPS 2018]

Reviewer 1



The paper proposes a nonparametric Bayesian generative model for jointly evaluating the local spectra corresponding to a Gaussian process latent signal, and shows how to explicitly obtain closed form expressions for the same owing to its gaussian structure. Even though I am not an expert in the field, to the best of my knowledge this is a new line of work. The paper is fairly readable . Bayesian non-parametric models have been readily used over the past couple of years to model in a component-free manner, and this line of work directly connects to that. The work appears significant since it opens up a plethora of ways to model signals using bayesian methods.

Reviewer 2



TITLE Bayesian Nonparametric Spectral Estimation PAPER SUMMARY The paper discusses Bayesian spectral estimation under a Gaussian process signal model, and presents a method for posterior inference of the spectral density. QUALITY Everything appears technically correct. CLARITY The paper is extremely well written and clear. ORIGINALITY To my knowledge, the methods presented in the paper have not been considered before. After reading the paper, the approach seems almost obvious. SIGNIFICANCE I think this contribution is very valuable, as it provides a fairly simple, direct, and consistent way to approach quantifying uncertainty in spectral estimation, which works in many cases including under non-uniform sampling. FURTHER COMMENTS Fast Fourier Transform (FFT) -> Do you mean discrete Fourier transform (DFT)? Perhaps you should make a distinction between the power spectra density and the energy spectral density. (rather than a parametric function) -> The AR method is also a stochastic process. Perhaps you mean to distinguish between parametric and non-parametric model based approaches Notice that the assumption of stationarity is required... -> I do not see why that is true Figure 1 is a bit confusing as it does not distinguish between stochastic and deterministic nodes Line 103: ...where the last two identities in each line are... -> where the last identity in each line is Figure 4: The rightmost subfigure is difficult to read. Edit: I have read the other reviews and the authors' response and remain confident that this paper should be a clear accept.

Reviewer 3



Spectral estimation is the task of finding the power associated with a signal across the frequency range of the signal. In classical signal processing applications, this is a solved problem when the time series is fully observed and equally spaced in time. However, in many cases, there are either missing observations, or the data is unevenly spaced in time. Here, probabilistic methods can be extremely valuable, as we can use nonparametric methods to provide better interpolation of the frequency domain, whilst handling the uncertainty in a principled way. This paper proposes a method to tackle this problem, with exact inference. A crucial component of this paper is windowing of the signal: without it, the Fourier transform of the GP would not exist. The authors here chose a version of the exponential window, which I presume is chosen for mathematical convenience rather than for any particular property (such as spectral leakage properties). From here, paper follows through the mechanics of deriving the local-spectrum GP using this windowing technique, the full derivation of which appears in Appendix 1.1. As I understand it, inference proceeds by sampling the time-domain GP, and then sampling in the spectral domain conditioned on the observations and given the time-domain functions. The final part is to compute the spectral energy, which is the (random) squared norm of the real and imaginary parts, which since they are independent GPs, results in a Chi-squared process. The rest of the paper is concerned with model consistency, and showing that the proposed model forms the limit of the popular Lomb-Scargle method (used for non-uniformly sampled data) as the number of sinusoids tends to infinity. Empirical analysis demonstrates that the proposed method is competitive with the Lomb-Scargle method, whilst being able to proved uncertainty estimates. Overall this is a well written paper, and provides significant contributions to the field of Bayesian spectral analysis. Specific comments: I find the use of the same K to represent the kernel function (of two variables) and the stationary kernel function (taking the difference between variables as input) confusing - perhaps $K_S$ could be used? When is K a function of 1 and two variables? (e.g. in 64 it is again used as $K(T)$). 71 “the signal and its spectrum are jointly Gaussian”. Is the spectrum Complex not Gaussian? What does it mean to say that they are jointly Gaussian? Eq 21, t on LHS is a vector 212 Fig 5.1 -> Fig 4? 233 what does “training” mean in the context of the Lomb-Scargle method? 252 available -> able Note: Thanks to the authors for the clarifications in the feedback.